# Unsupervised Machine Learning to Identify Patient Clusters and Tailor Perioperative Care in Colorectal Surgery

**DOI:** 10.3390/diagnostics15172124

**Published:** 2025-08-22

**Authors:** Philip Deslarzes, He Ayu Xu, Jean Louis Raisaro, Martin Hübner, Fabian Grass

**Affiliations:** 1Department of Visceral Surgery, Lausanne University Hospital CHUV, University of Lausanne (UNIL), 1015 Lausanne, Switzerland; philip.deslarzes@chuv.ch (P.D.); martin.hubner@chuv.ch (M.H.); 2Biomedical Data Science Center, Lausanne University Hospital CHUV, University of Lausanne (UNIL), 1015 Lausanne, Switzerland; he.xu@chuv.ch (H.A.X.); jean.raisaro@chuv.ch (J.L.R.)

**Keywords:** clusters, ERAS, cluster transition, outcomes, colorectal surgery, recovery

## Abstract

**Background**: The aim of the present study was to apply machine learning (ML) techniques to define clusters relating patient demographics, compliance, and outcome variables in colorectal enhanced recovery after surgery (ERAS) patients and improve data-driven, predictive decision-making. **Methods**: To uncover inherent patient subgroups from the data without pre-defined labels, the unsupervised K-means clustering algorithm was utilized. This technique was selected for its effectiveness in partitioning patients into distinct groups by iteratively assigning them to the nearest cluster mean, thereby minimizing within-cluster variance across key variables. The top five recovery goals and the top 10 clinical outcome variables were defined based on clinical considerations (incidence and importance). In a second step, the cluster transition was traced by monitoring the transitions between clusters from demographic through compliance to outcome variables. **Results**: A total of 1381 patients were available for final analysis, revealing three clusters (low risk, *n* = 490, 36%; intermediate risk, *n* = 157, 11%; and high risk, *n* = 734, 53%) for demographic, two clusters (high compliance, *n* = 1011, 73%, and low compliance *n* = 370, 27%) for perioperative, and two clusters (good and poor outcomes) for the top five recovery goals and the top 10 clinical outcomes, respectively. The cluster transition for the top five recovery goals and the top 10 clinical outcomes revealed that most patients (488/490, 99.6%) of the low-risk demographic cluster had high perioperative compliance, and over 90% of them had favorable functional and clinical outcomes. Of the 2/3 of intermediate risk patients who had poor perioperative compliance, over 40% had a poor functional recovery, whereas 83% had good clinical outcomes. Of the high-risk demographic group, 100% (734/734) had low perioperative compliance, and over 40% of them had poor functional recovery. **Conclusions**: This ML-based analysis of demographic, compliance, and recovery clusters and associated cluster transition allowed us to identify patient clusters as a first step to tailored ERAS protocols aiming to improve compliance and outcomes.

## 1. Introduction

Standardized enhanced recovery after surgery (ERAS) pathways help to foster best evidence-based practice into clinical routine [1]. The current ERAS protocol for colorectal surgery includes over 20 individual care items with a focus on restrictive use of perioperative fluids, multimodal pain management, early postoperative nutrition, and mobilization. ERAS is a standardized and validated concept aiming to minimize surgical stress, maintain physiological stability, and promote postoperative recovery [2,3]. The success of ERAS strongly correlates with compliance to the care bundle in a dose (high compliance)–effect (better outcomes) pattern [1,4]. Currently, one perioperative protocol applies to all patients, regardless of comorbidities, surgical difficulty, or expected recovery. Since postoperative complications are strongly associated with individual patient characteristics and the surgical setting, there may be a need to refine protocols towards a more personalized, tailored approach [5]. Machine learning (ML) techniques, which have emerged recently, allow for the processing of complex and high-dimensional data to uncover non-linear correlations and may help to refine and adapt care protocols from a large-scale perspective [6]. These techniques provide a quality improvement opportunity within the ERAS concept to improve data-driven, predictive decision-making [7].

To refine and individualize the presently universal ERAS program, the aim of the present study was to apply ML techniques to define clusters in colorectal ERAS patients for tailored perioperative care. In a second step, the cluster transition was analyzed, aiming to tailor perioperative care for the best possible outcomes.

## 2. Methods

This is a monocentric retrospective cohort study including consecutive patients who underwent elective colorectal surgery in the visceral surgery department at CHUV between 1st May 2011 and 31st December 2022.

The ERAS protocol was implemented in May 2011 and considered standard of care, and no patient was excluded from the pathway [8,9]. Our institutional ERAS protocol followed the published ERAS society guidelines [2,10], including clinical pathways, data management, audit, and logistics. Since implementation, all patients undergoing colorectal surgery were prospectively included in the institutional ERAS database with periodic database audits every 2-4 weeks. A dedicated ERAS nurse collected patient demographics, surgical details, complications, length of stay, and compliance items; a total of 135 variables/patient were collected (Appendix A, online). No patient was excluded from the analysis, providing general consent for research purposes was available. This study was approved by the institutional review board (CER-VD-2022.01443).

The collected variables were retrieved from the ERAS Interactive Audit System (EIAS), providing clear definitions for all variables [11,12,13]. Only a few items were excluded for being unsuitable. For the purpose of this study, data items were assigned to demographic, compliance, and outcome variables according to previous institutional publications [14,15]. The individual variables are detailed in Appendix A (online).

### 2.1. Statistical Analysis

All variables were preprocessed to improve the quality of later analysis. The preprocess included imputing missing variables using the mean or median based on the nature of the variable, standardizing variables using the z-score, and converting categorical variables (such as sex) using one-hot encoding.

Given the presence of both numerical and categorical variables, the unsupervised K-means algorithm was used for patient clustering (Python scikit-learn package version 1.4.2) [16].

To prepare the data for clustering and enhance the performance of the algorithm, we first performed Principal Component Analysis (PCA) on each variable group. This dimensionality reduction technique was implemented for two primary reasons: first, to reduce statistical noise by removing less-informative components, and second, to mitigate the “curse of dimensionality,” where an excessive number of features can degrade the performance of clustering algorithms like K-means. For each analysis, we retained the minimum number of principal components required to explain at least 75% of the total variance in the data.

Following this preprocessing step, the resulting principal components were used as input for the unsupervised K-means algorithm (Python scikit-learn package version 1.4.2) to perform patient clustering. The key hyperparameter for the model was the number of clusters, which was tested for values of 2, 3, 4, and 5. The optimal number of clusters for each variable group was determined by selecting the value that yielded the highest Silhouette score, indicating the best-defined and most distinct clusters (see the Appendix A for more detail).

Clustering analysis was conducted separately for demographic variables, compliance variables (perioperative variables), and outcome variables, as well as for the combination of demographic and perioperative variables.

The top 5 recovery goals and the top 10 clinical outcome variables were defined based on clinical considerations (incidence and importance). This selection was necessary to perform the clustering analysis and to study the cluster transition throughout the perioperative journey. The top 5 recovery goals included time to tolerate solid food, total intravenous volume of fluids on postoperative day (POD) 0, mobilization > 6 h on POD 2, delta weight (weight change) on POD 2, and weight change > 2.5 kg on POD 2. The 10 top clinical outcomes were major complications (Clavien ≥ 3b), respiratory complications, infectious complications, renal dysfunction, anastomotic leak, nasogastric tube (NGT) reinsertion, death, reoperation, pain on POD 1, and ileus, according to previously published definitions [17].

To identify significant differences between the identified clusters, a series of statistical tests was conducted. For the three demographic clusters, differences in numerical variables were assessed using one-way Analysis of Variance (ANOVA), while differences in categorical variables were assessed using the chi-square test (Table 1). For all pairwise comparisons, independent *t*-tests were employed to examine differences between the two perioperative clusters (Table 2) and between the clusters distinguished by selected recovery goals and clinical outcomes (Table 3).

### 2.2. Outcomes

The primary study aim was the identification of patient clusters, using unsupervised K-means algorithms, who perform differently in the perioperative journey regarding demographics, perioperative ERAS compliance, and outcomes. Clusters were then linked among them to predict a perioperative journey for a given patient within a given cluster, helping to identify preventive measures to potentially improve outcomes.

In a second step, patient trajectories were mapped by quantifying the transitions between clusters at sequential time points. The analytical process involved two main steps:
Preoperative to perioperative transition: We first quantified patient transitions from the three initial demographic (preoperative) clusters to two summary clusters derived from a K-means analysis of combined demographic and perioperative variables.Perioperative to postoperative transition: Next, we analyzed the transitions from the combined preoperative and perioperative clusters to the final postoperative clusters. This step was performed twice, as the postoperative clusters were independently generated based on two distinct sets of variables: (a) selected recovery goals and (b) clinical outcomes.

The observed transition frequencies and pathways were then used to classify patients into low-, medium-, or high-risk trajectories.

## 3. Results

In total, 1732 consecutive patients underwent elective colorectal surgery over the study period. After applying exclusion criteria (no research consent), 1381 patients were available for final analysis.

### 3.1. Cluster Analysis

After performing K-means analysis on the three groups of variables, three clusters for demographic variables, two clusters for perioperative variables, and three clusters for postoperative variables were identified (Figure 1). Two clusters were identified when focusing on the top five recovery goals and the top 10 clinical outcomes, respectively.

The three clusters for demographic variables were as follows: cluster 1 (low-risk cluster, *n* = 490, 36%), cluster 2 (intermediate risk cluster, *n* = 157, 11%), and cluster 3 (high-risk cluster, *n* = 734, 53%, Table 1). The three demographic clusters did not show significant differences regarding age, weight change, height, BMI, gender, smoking, and diabetes status. However, alcohol consumption, severe heart disease, and severe pulmonary disease differed significantly between the clusters.

Two different clusters for postoperative recovery (top five) were identified: cluster 1 (*n* = 1012, 73%, corresponding to prompt recovery) and cluster 2 (*n* = 369, 27%, corresponding to delayed recovery), with respective differences in the top five recovery items (Table 3a). Clinical outcomes (top 10) were grouped into two distinct clusters: cluster 1 (good outcomes, *n* = 1245, 90%) and cluster 2 (poor outcomes, *n* = 136, 10%), with significant differences regarding all top 10 outcomes (Table 3b). More detailed analysis of the overall postoperative variables (beyond the predefined top five recovery goals and the top 10 clinical outcome items) is displayed in Appendix A (online), revealing three different clusters with good (*n* = 667, 49%), intermediate (*n* = 535, 39%), and poor (*n* = 162, 12%) outcomes.

Clustering analysis for (a) pre-, (b) peri-, and (c) postoperative variables was conducted. T-tests were conducted between the two identified clusters for both groups, and the mean values of each cluster are displayed in Table 2. The scatter plots of clustered variables are based on principal components 1 and 2 after Principal Component Analysis transformation. The data points are color-coded and displayed according to their assigned cluster:(a)Clusters for low- (green), intermediate- (orange), and high- (blue) risk groups.(b)Clusters for low (orange) and high (green) compliance.(c)Clusters for good (green) and poor (orange/blue) outcomes.

### 3.2. Cluster Transition

The cluster transition for the top five recovery goals and the top 10 clinical outcomes is displayed in Figure 2a and Figure 2b, respectively. Almost all patients (488/490, 99.6%) of the low-risk demographic cluster had high perioperative compliance, and over 90% (464/488 and 452/488, respectively) of them had both favorable functional (recovery) and clinical outcomes. The analysis further revealed that 1/3 of patients in the intermediate risk group had high perioperative compliance, with again, over 90% (50/51 and 47/51) of them presenting both favorable functional (recovery) and clinical outcomes. Of the 2/3 of intermediate risk patients who had poor perioperative compliance, over 40% presented with delayed functional recovery, whereas 83% had encouraging clinical outcomes. Of the high-risk demographic group, 100% (734/734) had low perioperative compliance, and over 40% of them had poor functional recovery, whereas 89% still resulted in the good clinical outcome cluster.

## 4. Discussion

The present cluster analysis of a large institutional colorectal ERAS dataset revealed the specific cluster transition related to different demographic and compliance patterns, ultimately resulting in either encouraging or poor functional and clinical outcomes. This quality improvement initiative may be a first step to innovate “one size fits all” ERAS protocols into a more individualized and patient-centered approach, providing opportunities to anticipate and modify arduous cluster transition and improve functional and clinical recovery.

This study analyzed patient characteristics through an unsupervised ML methodology, leaving aside intuitive or “to be expected” risk constellations and thus aiming to acquire new knowledge of the perioperative patient journey. The choice of K-means was a pragmatic decision that balanced model assumptions, research objectives, and the need for clear and interpretable outcomes. Postoperative outcome analysis initially revealed three clusters, which were intentionally narrowed down to the clinically most relevant top recovery goals and clinical outcomes to facilitate analysis of the cluster transition. Similar approaches of ML-based cluster analyses have been carried out on multiple clinical metrics in different surgical fields [18,19,20]. However, in the field of perioperative ERAS care, a more individualized approach has not yet been described. The longstanding ERAS experience of our center with a large, prospective, and regularly audited 10-year institutional dataset provided an ideal framework for such an analysis. Despite an established ERAS protocol in our institution, postoperative complications such as ileus and pulmonary complications or mobilization patterns have been extensively studied and revealed multiple areas of improvement [14,17,21]. While our efforts initially focused on stringent perioperative fluid management and nutrition to face these challenges [22,23], the Leuven group decided to focus on key components to increase protocol adherence [24].

Demographic clusters revealed a majority (53%) of high-risk patients in our institution, which is related to its role as a tertiary referral center. Interestingly, these patients consistently showed poor compliance patterns to perioperative care and hence accumulated important risk factors for adverse postoperative outcomes in both pre- and intraoperative categories, ultimately resulting in delayed postoperative recovery in over 40% of patients. In consequence, preoperative knowledge of a cluster 3 (= high-risk) patient should trigger efforts to optimize modifiable demographic risk factors prior to surgery through prehabilitation efforts (potentially allowing a shift to cluster 2) in order to increase perioperative compliance in the sense of a tailored pathway. This may ultimately contribute to better clinical and functional outcomes.

On the other side of the spectrum, preoperative cluster 1 (= low risk) patients, representing 36% of the cohort, were highly compliant with the ERAS protocol and had both uneventful functional and clinical recovery in over 90% of cases, including the few patients with low perioperative compliance. These patients appear predestined to a prompt and smooth recovery, and efforts can be safely reallocated to the more fragile subset of patients [25]. Furthermore, these patients may qualify for slim ultra-ERAS protocols and colectomy in an outpatient setting, provided logistic prerequisites are available and strict eligibility criteria are applied [26,27,28]. The 7% of low-risk patients with poor clinical outcomes are hence most likely linked to surgical complications, in line with what can be reasonably expected in the recent surgical literature for elective colorectal surgery [29]. From a more general standpoint, the present study suggests that perioperative care and compliance primarily impact functional recovery rather than clinical outcomes, which may more directly depend on the surgical procedure.

The intermediate risk group is interesting from several standpoints. First, two-thirds belonged to the low-compliance group, emphasizing the importance of investing in adapted care protocols for improved compliance, especially considering that over 40% of these low-compliance group patients struggled with postoperative recovery and had a >20% higher risk for adverse clinical outcomes than patients with high compliance. Second, this group may benefit the most from an adapted care protocol with intensified care tailored to individual patient needs. While this present study focused on the big picture of patient clusters and trajectories, a more detailed analysis of patient characteristics and risk profiles for frequent adverse outcomes is needed to refine the ERAS protocol for the specific and unique surgical patient. Taken together, full investment in ERAS is of particular importance in this intermediate risk group and underpins the importance of aiming for high compliance with optimal perioperative care. Figure 3 highlights areas of improvement for optimized cluster transition.

This study has limitations related to a large and heterogeneous single-center dataset of all-comers and its focus on patient clusters and trajectories from a more general standpoint rather than an in-depth analysis of sensitive outcomes and patient profiles. Furthermore, the list of confounders is not exhaustive. The follow-up study will focus on risk prediction models to refine the “one size fits all” ERAS protocol. Further large-scale studies in multicenter settings are then needed to confirm, validate, and generalize the findings of our single-center experience for extrapolation to other institutions and settings.

In conclusion, this ML-based analysis of demographic, compliance, and recovery clusters has the potential to optimally prepare patients for their perioperative colorectal surgery journey by customizing individual care to specific needs according to risk profiles within the ERAS pathway, ultimately allowing better resource allocation and targeted care of more vulnerable patients. Future studies should validate these preliminary findings in adequately designed multicenter settings to increase generalizability and further improve machine learning models for perioperative care.

## Figures and Tables

**Figure 1 diagnostics-15-02124-f001:**
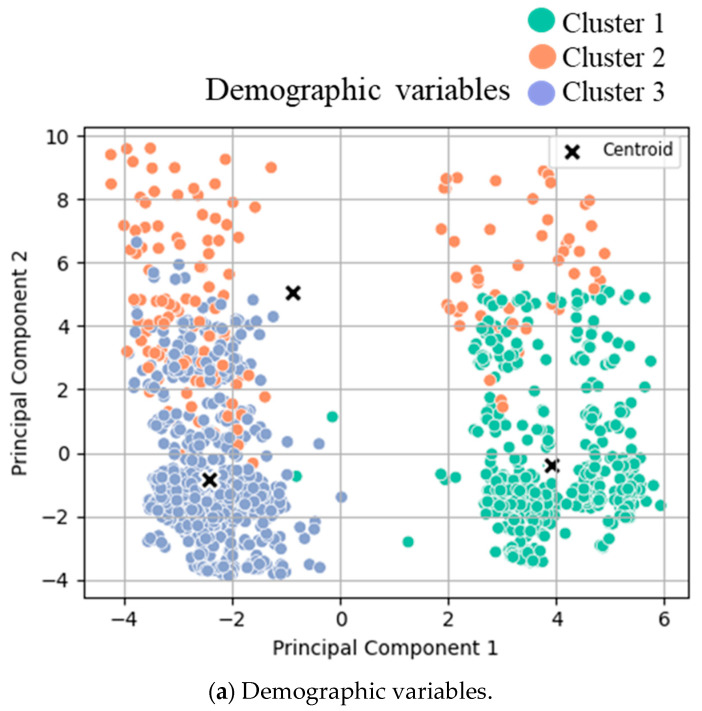
Cluster analysis visualization. K-means clustering results with the best number of clusters (selected by the Silhouette score) on PCA-reduced demographic data with the first and second principal components. Each point represents a data sample colored by its assigned cluster.

**Figure 2 diagnostics-15-02124-f002:**
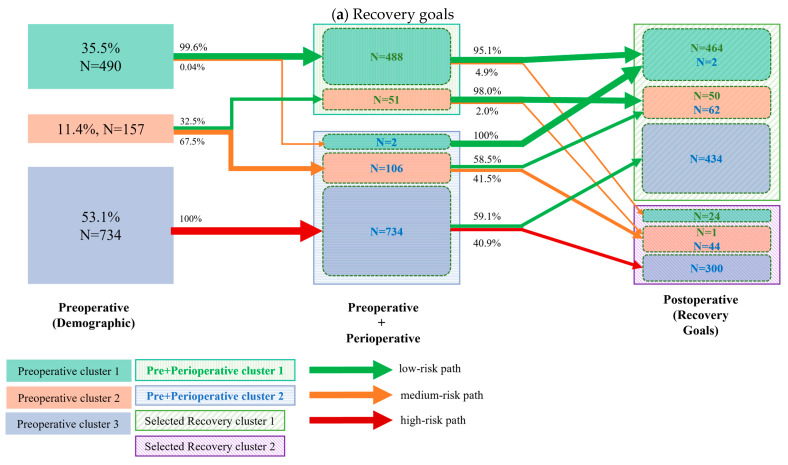
Cluster transition: The left column represents the clusters (different color boxes) identified from demographic variables (also referred to as preoperative variables); the number and percentage of patients per cluster are presented in each cluster block. The middle column represents the cluster identified by combining both preoperative (demographic) and perioperative variables (marked as the Pre + Perioperative cluster). The clusters are marked in shadowed boxes. Within each Pre+Perioperative cluster, the colored boxes indicate how many patients from the preoperative cluster are transitioned to the current cluster. The right column represents the clusters identified by the selected recovery goals. The two identified clusters are marked by the shadowed box. The solid boxes within the identified clusters indicate the number of patients transitioned from the preoperative cluster, while the *N* values of different colors indicate the number of transitioned patients from the previous cluster. The arrows between clusters indicate the percentage of transitioned patients. The colors of the arrows indicate the risk level of the transitioned patients. The cluster transition according to pre- and perioperative clusters regarding (**a**) the top 5 recovery goals and (**b**) the top 10 clinical outcomes. Arrow thickness corresponds to the percentage of patients within a cluster transition.

**Figure 3 diagnostics-15-02124-f003:**
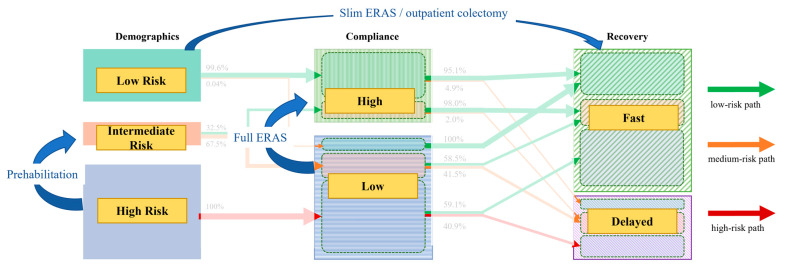
Areas of improvement. Cluster transition diagram highlighting preventive measures and consequences to address different pathways. ERAS—enhanced recovery after surgery.

**Table 1 diagnostics-15-02124-t001:** Demographic variables by cluster.

Variable (Numeric)	Cluster 1 (*n* = 490)	Cluster 2 (*n* = 157)	Cluster 3 (*n* = 734)	*p*-Value	Sample Mean (*n* = 1381)
Age (years)	61.2	63.2	60.8	0.24	61.2
Weight 6 m prior to admission (kg)	74	74.9	75	0.36	74.6
Preoperative body weight (kg)	73.3	73.2	74.4	0.46	73.8
Preoperative weight change (kg)	−1.1	−1.3	−1.1	0.85	−1.1
Height (cm)	168.7	168.2	169.2	0.46	168.9
BMI (kg/m^2^)	25.6	25.8	26	0.58	25.8
Length of incision (cm)	13.1	13.1	12.7	0.72	12.9
Variable (Categorical)	Cluster 1 (*n* = 490)	Cluster 2 (*n* = 157)		Cluster 3 (*n* = 734)	*p*-Value
Gender (male)	57.9%	50.9%		58%	0.24
Non-smoker or stopped	81.3%	76.4%		75.2%	0.13
Excess alcohol ingestion	4.5%	7.6%		15.9%	<0.001
Diabetes mellitus	8.9%	15.3%		11.8%	0.13
Severe heart disease	2%	6.4%		9.6%	<0.001
Severe pulmonary disease	0.4%	3.2%		4.1%	<0.001

Clustering results for preoperative variables. For numerical variables, ANOVA tests were conducted between the 3 identified clusters, and the mean value of each cluster is displayed. For the categorical variable, chi-square tests were conducted between the clusters, and the percentages of the values are shown. m—months; BMI—body mass index. Cluster 1—low-risk cluster; Cluster 2—intermediate risk cluster; and Cluster 3—high-risk cluster.

**Table 2 diagnostics-15-02124-t002:** Perioperative variables by cluster.

Variable	Cluster 1 (*n* = 1011)	Cluster 2 (*n* = 370)	*p*-Value
Core body temperature at end of operation (°C)	36.3	36.4	<0.0001
IV volume of crystalloids, intraoperatively (mL)	1310	2520	<0.0001
IV volume of colloids, intraoperatively (mL)	80	400	<0.0001
Total IV volume of fluids, intraoperatively (mL)	1380	2980	<0.0001
Total IV volume of fluids on day zero (mL)	2160	4230	<0.0001
Morning weight (kg)			
- On POD 1	74	78.8	<0.0001
Weight change POD 1 (kg)	0.8	2.1	<0.0001
Morning weight (kg)			
- On POD 2	73.1	80.4	<0.0001
Weight change POD 2 (kg)	0.8	2.7	<0.0001
Morning weight (kg)			
- On POD 3	72.6	80.5	<0.0001
Weight change POD 3 (kg)	0.5	2.5	<0.0001
Oral fluids, total volume taken (mL)			
- On POD 0	1010	660	<0.0001
- On POD 1	1620	1350	<0.0001
- On POD 2	1600	1350	<0.0001
Oral nutritional supplements, energy intake (kcal)			
POD 0	130	50	<0.0001
POD 1	350	190	<0.0001
POD 2	310	210	<0.0001
POD 3	180	120	0.0001

Clustering results for perioperative variables. T-tests were conducted between the 2 identified clusters, and the mean value of each cluster is displayed. IV—intravenous; POD—postoperative day. Cluster 1—high compliance; Cluster 2—low compliance.

**Table 3 diagnostics-15-02124-t003:** Postoperative outcomes by cluster.

(a) Recovery goals				
*Recovery Item*	Cluster 1 (*n* = 1012)	Cluster 2 (*n* = 369)	*p*-Value	Sample Mean (*n* = 1381)
Time to tolerate solid food (nights)	2.4	4	<0.0001	2.8
Total IV volume POD 0	2570	3110	<0.0001	2710
Mobilization > 6 h on POD 2	59.2%	1.1%	<0.0001	603
Weight change POD 2 (kg)	1.36	1.27	0.45	1.34
Weight change > 2.5 kg POD 2	21.7%	59.1%	<0.0001	278
(b) Clinical outcomes				
*Type of Complication*	Cluster 1 (*n* = 1245)	Cluster 2 (*n* = 136)	*p*-Value	Sample Mean (*n* = 1381)
Major complication	1%	62.5%	<0.0001	403
Respiratory complication	2.8%	33.1%	<0.0001	80
Infectious complication	7.6%	64.7%	<0.0001	183
Renal dysfunction	0.6%	2.2%	<0.0001	72
Anastomotic leak	0.1%	16.8%	<0.0001	88
Delayed first passage of stool > POD 3	7.6%	44.1%	<0.0001	155
Death	0.1%	5.9%	<0.0001	9
Reoperation	1.2%	80.1%	<0.0001	124
Pain on POD 1 (VAS)	3.8	4.2	0.08	3.82
Ileus	6.8%	30.9%	<0.0001	127

Clustering results for (a) the top 5 recovery goals and (b) the top 10 clinical outcomes. T-tests were conducted between the 2 identified clusters for both groups, and the mean values of each cluster are displayed. IV—intravenous; POD—postoperative day; and VAS—visual analogue scale. (a) Cluster 1: fast recovery; Cluster 2: delayed recovery. (b) Cluster 1: good outcomes; Cluster 2: poor outcomes.

## Data Availability

The data will be made available upon request.

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
