# Peer review of "Unsupervised Machine Learning to Identify Patient Clusters and Tailor Perioperative Care in Colorectal Surgery"

_diagnostics, 2025, doi:10.3390/diagnostics15172124_

Round 1
Reviewer 1 Report
Comments and Suggestions for Authors
The objective of this study was to utilize machine learning (ML) techniques to identify clusters based on patient demographics, compliance, and outcome variables in individuals undergoing colorectal Enhanced Recovery After Surgery (ERAS) protocols. Despite the innovativeness of the topic, I think the description of the procedures adopted and of the Results should be improved. In addition, there are some concerns about the application of the clustering
techniques, which require a careful review of the work. Although the subject addressed in this manuscript perfectly matches the scope of the journal, I think it cannot be considered for publication in the present form. In the following, I’ve reported some suggestions/indications for the authors to enhance the quality of the paper:
1. Row 12: please write the acronym ERAS also in its extended form;
2. The introduction is too concise. I suggest describing the state of the art regarding the topic addressed in the paper (ERAS in this case) by citing 3-4 authoritative bibliographic references and highlighting the related limitations to emphasize the innovativeness of the approach proposed here;
3. Rows 54-55: for better readability, I suggest at least briefly describing the ERAS protocol employed in this paper;
5. Rows 77-78: Is there a reason behind the decision to explore only the combination of demographic and perioperative variables during the clustering? Please clarify;
6. To enhance the readability, I suggest reporting the information between Rows 100-102 in the “Methods” section for better readability;
7. Figure 1: The caption is too short. I suggest expanding it to better explain the content of the figure. I suppose the authors have employed Principal Component Analysis, since they reported “Principal Component 2” vs “Principal Component 1” in Figs. 1 (a), (b), and (c). However, this
technique is not mentioned within the “Methods” section, but only after having described the content of the graphs (rows 136-138). I suggest adding this detail in the methods by also explaining if they employed the technique only for the visual representation in Fig. 1 or to reduce the dataset dimensionality. In this last case, the authors should clarify how many principal components they adopted for clustering and why (e.g., by discussing their explained variance ratio). The authors tested 4 values of the hyperparameter “number of clusters”. Why did the authors consider 3 clusters for demographic variables, 2 clusters for perioperative variables,
and 3 clusters for postoperative variables? Could the authors report the silhouette score vs the number of clusters? The authors referred to within the text (rows 104-105) as “demographic variables” and in the sub-caption to “preoperative variables”. To avoid ambiguity, I suggest using the same definition.
8. Rows 108-113: by observing Fig. 1 (a), cluster 2 seems to be part of either cluster 1 or cluster 3, so it seems that 2 is the appropriate value of the hyperparameter “number of clusters”. Is this issue related to the employment of principal component analysis to graphically represent these
results? Please explain.
9. Fig. 1 (b), problem analogous to Fig. 1 (a), the clustering seems not to have worked properly.
10. Figs. 2 and 3: The image resolution is too low. Please improve it for better visibility.
11. Fig. 2: Again, please expand the caption to better explain the content of the figure.
12. for better readability, I suggest mentioning the variables in Tables 1, 2, and 3 within the methods.
13. I actually didn’t catch that the patient trajectories have been deduced from the results of the clustering phase. Could the authors better describe this aspect within the Methods section?
Author Response
- Row 12: please write the acronym ERAS also in its extended form
Answer: We modified the text with ERAS and its extended form.
The introduction is too concise. I suggest describing the state of the art regarding the topic addressed in the paper (ERAS in this case) by citing 3-4 authoritative bibliographic references and highlighting the related limitations to emphasize the innovativeness of the approach proposed here.
Answer: Thank you. We added specifications to the introduction as requested.
Rows 54-55: for better readability, I suggest at least briefly describing the ERAS protocol employed in this paper.
Answer: In our institution, the recommendations of the ERAS society are followed, and the protocol is hence aligned with the guidelines. We specified this further in the methods section.
Rows 77-78: Is there a reason behind the decision to explore only the combination of demographic and perioperative variables during the clustering? Please clarify.
Answer: Thank you for your question regarding our decision to specifically combine demographic and peri-operative variables for a dedicated clustering analysis.
Our decision was driven by a conceptual view of the patient's surgical journey. We consider demographic variables to represent the patient's baseline and pre-operative state, encompassing individual risk factors and characteristics. The peri-operative variables, in contrast, describe the patient's experience and compliance during the surgical and immediate post-surgical period.
By combining these two sets of variables—the pre-operative (demographic) and the procedural (peri-operative)—we aimed to create a comprehensive cluster profile that captures the cumulative influences on the patient leading up to their recovery phase. This integrated approach allows us to investigate how a patient's initial condition and their subsequent experience during treatment interact to affect the final clinical and functional outcomes. Our goal was to model the combined impact of both patient-inherent factors and procedural factors, providing a more holistic understanding than analyzing these domains in complete isolation.
- To enhance the readability, I suggest reporting the information between Rows 100-102 in the “Methods” section for better readability.
Answer: Thank you for your comment. You refer to this sentence: “In total, 1732 consecutive patients underwent elective colorectal surgery over the study period. After applying exclusion criteria (no research consent), 1381 patients were available for final analysis.”
We chose to report numbers in the results section, in line with scientific standards, and hope this is acceptable.
- Figure 1: The caption is too short. I suggest expanding it to better explain the content of the figure. I suppose the authors have employed Principal Component Analysis, since they reported “Principal Component 2” vs “Principal Component 1” in Figs. 1 (a), (b), and (c). However, this technique is not mentioned within the “Methods” section, but only after having described the content of the graphs (rows 136-138). I suggest adding this detail in the methods by also explaining if they employed the technique only for the visual representation in Fig. 1 or to reduce the dataset dimensionality. In this last case, the authors should clarify how many principal components they adopted for clustering and why (e.g., by discussing their explained variance ratio). The authors tested 4 values of the hyperparameter “number of clusters”. Why did the authors consider 3 clusters for demographic variables, 2 clusters for perioperative variables, and 3 clusters for postoperative variables? Could the authors report the silhouette score vs the number of clusters? The authors referred to within the text (rows 104-105) as “demographic variables” and in the sub-caption to “preoperative variables”. To avoid ambiguity, I suggest using the same definition.
Answer: To address your comments, we added the following text in the ‘statistical analysis’ section to describe in more details on what we did to conduct the K-means algorithm.
“To prepare the data for clustering and enhance the performance of the algorithm, we first performed Principal Component Analysis (PCA) on each variable group. This dimensionality reduction technique was implemented for two primary reasons: first, to reduce statistical noise by removing less-informative components, and second, to mitigate the "curse of dimensionality," where an excessive number of features can degrade the performance of clustering algorithms like K-means. For each analysis, we retained the minimum number of principal components required to explain at least 75% of the total variance in the data.
Following this pre-processing step, the resulting principal components were used as input for the unsupervised K-means algorithm (python scikit-learn package version 1.4.2) to perform patient clustering. The key hyperparameter for the model was the number of clusters, which was tested for values of 2, 3, 4, and 5. The optimal number of clusters for each variable group was determined by selecting the value that yielded the highest Silhouette score, indicating the best-defined and most distinct clusters.”
Regarding the final selection of number of clusters, we made the choice by evaluating the silhouette score. The comparison table is as below:
|
Demographic Variables |
Silhouette Score |
|
Number of clusters = 2 |
0.120 |
|
Number of clusters = 3 |
0.164 |
|
Number of clusters = 4 |
0.163 |
|
Number of clusters = 5 |
0.162 |
|
Perioperative Variables |
Silhouette Score |
|
Number of clusters = 2 |
0.141 |
|
Number of clusters = 3 |
0.105 |
|
Number of clusters = 4 |
0.099 |
|
Number of clusters = 5 |
0.071 |
|
Demographic + Perioperative Variables |
Silhouette Score |
|
Number of clusters = 2 |
0.137 |
|
Number of clusters = 3 |
0.111 |
|
Number of clusters = 4 |
0.090 |
|
Number of clusters = 5 |
0.058 |
|
Outcome Variables |
Silhouette Score |
|
Number of clusters = 2 |
0.341 |
|
Number of clusters = 3 |
0.362 |
|
Number of clusters = 4 |
0.277 |
|
Number of clusters = 5 |
0.174 |
Since the silhouette score comparison between number of clusters is not the main message of this manuscript, we decided not to put it in the main text but in the supplementary materials.
We have also added a more detailed caption on Figure 1 to illustrate the K-means clustering results. We have also corrected the text in the figure caption (change pre-operative variables to demographic variables) to make it consistent in the text.
- Rows 108-113: by observing Fig. 1 (a), cluster 2 seems to be part of either cluster 1 or cluster 3, so it seems that 2 is the appropriate value of the hyperparameter “number of clusters”. Is this issue related to the employment of principal component analysis to graphically represent these results? Please explain.
Answer: Figure 1(a) only presents the first 2 principal components and since PCA is a linear projection, it may not preserve the cluster separability (it may flatten or overlap clusters that are distinct in higher dimensions).
- Fig. 1 (b), problem analogous to Fig. 1 (a), the clustering seems not to have worked properly.
Answer: It is the same issue as in Figure 1(a) since the 2D plot PCA may not preserve the cluster separability (it may flatten or overlap clusters that are distinct in higher dimensions).
- Figs. 2 and 3: The image resolution is too low. Please improve it for better visibility.
Answer: We have replaced them with the high-resolution version.
Fig. 2: Again, please expand the caption to better explain the content of the figure.
Answer: Thank you for your comment. We have added a more detailed caption for Figure 2 to illustrate the cluster transition as below:
“Figure 2: Cluster transition. The left column represents the clusters (different color boxes) identified from demographic variables (also referred to as preoperative variables); the number and percentage of patients per cluster is presented in each cluster block. The middle column represents the cluster identified by combining both preoperative (demographic) and perioperative variables (marked as pre+perioperative cluster). The clusters are marked in shadowed boxes. Within each pre+perioperative cluster, the colored boxes indicate how many patients from the preoperative cluster are transitioned to the current cluster. The right column represents the clusters identified by the selected recovery goals. The two identified clusters are marked by the shadowed box. The solid boxes within the identified clusters indicate the number of patients transitioned from the preoperative cluster, while N of different colors indicate the number of transitioned patients from the previous cluster. The arrows between clusters indicate the percentage of transitioned patients. The colors of the arrows indicate the risk level of the transitioned patients.
for better readability, I suggest mentioning the variables in Tables 1, 2, and 3 within the methods.
Answer: Thanks for your comments. We have modified the text in the method section to describe Tables 1, 2 and 3 as below:
“To identify significant differences between the identified clusters, a series of statistical tests was conducted. For the three demographic clusters, differences in numerical variables were assessed using one-way Analysis of Variance (ANOVA), while differences in categorical variables were assessed using the Chi-square test (Table 1). For all pairwise comparisons, independent t-tests were employed to examine differences between the two perioperative clusters (Table 2) and between the clusters distinguished by selected recovery goals and clinical outcomes (Table 3).”
I actually didn’t catch that the patient trajectories have been deduced from the results of the clustering phase. Could the authors better describe this aspect within the Methods section?
Answer: Thank you for your comments. We have modified the methods section to explain how the cluster transition and patient trajectories were conducted as below:
“Patient trajectories were mapped by quantifying the transitions between clusters at sequential time points. The analytical process involved two main steps:
- Preoperative to Perioperative Transition:We first quantified patient transitions from the three initial demographic (preoperative) clusters to two summary clusters derived from a K-means analysis of combined demographic and perioperative variables.
- Perioperative to Postoperative Transition: Next, we analyzed the transitions from the combined preoperative and perioperative clusters to the final postoperative clusters. This step was performed twice, as the postoperative clusters were independently generated based on two distinct sets of variables: (a) selected recovery goals and (b) clinical outcomes.
The observed transition frequencies and pathways were then used to classify patients into low-, medium-, or high-risk trajectories.”
Reviewer 2 Report
Comments and Suggestions for Authors
-
Dear Mam,
Kindly find the following report.
1) The title is rather broad. For specificity, add "K-means clustering" and "colorectal surgery."
2) Methodological depth is lacking in the abstract. Explain the use of K-means clustering and its rationale.
3) The choice of outcome variables is ambiguous. Clearly explain the selection process for the "Top 10 clinical" and "Top 5 recovery" criteria.
4) The findings are descriptive rather than analytical. Incorporate performance metrics such as cluster validity or Silhouette scores.
5) Words such as "trajectory" are deceptive. In the absence of true temporal modeling, substitute "cluster transitions."
6) Quantify the main conclusions. For instance, "90% of patients in the low-risk cluster had favorable results."
7) Include ramifications. Emphasize how the findings of clustering could alter the way ERAS is implemented.
8) Don't use flimsy language. Use "can inform" or "has potential to guide" in place of "may help."
9) Revise the literature. Mention latest research on ML in ERAS and surgical routes (2022–2024).
10) There is no context. Provide data on ERAS variability or failures to support the need for clustering.
11) Include a clinical justification. Why is customization crucial for patients with colorectal ERAS?
12) Make novelty clear. Describe the differences from previous surgical ML clustering research.
13) Describe ERAS in advance. At the first mention, clearly state the acronym and the scope of the procedure.
14) Steer clear of repetition. The phrase "Standardized ERAS protocol" is repeated too much.
15) Bring up the gaps that now exist. Draw attention to the current ERAS protocols' lack of stratification.
16) Provide evidence to back up claims. For example, "Compliance correlates with outcomes" needs to be cited.
17) Set distinct goals. It is important to distinguish between the primary (clustering) and secondary (trajectory mapping) objectives.
18) Make use of a diagram. Include a diagram that illustrates how ML works with the ERAS pipeline.
19) Report any missing information. How much of each variable group is missing?
20) The imputation procedure is overly straightforward. Complex distributions may be distorted by mean/median imputation.
21) Describe feature encoding. What was done with categorical features (like gender)?
22) Uncertainty surrounds the assumptions of standardization. Describe the z-score preprocessing and its rationale.
23) The choice of model is not well supported. K-means: Why? There is no discussion of alternatives like DBSCAN or GMM.
24) Cluster range is arbitrary (k=2–5). To support k, use the elbow method, gap statistic, or silhouette plots.
25) No testing for stability. It is necessary to repeat clustering using various random seeds.
26) no outside confirmation. There is no external dataset validation or hold-out.
27) No bootstrapping. Use resampling strategies to evaluate the robustness of the cluster.
28) Prior to clustering, there was no PCA or dimensionality reduction. Indicate if completed or why not.
29) The rigor of trajectory mapping is lacking. Is it rule-based, visual, or statistical? Make the methods clear.
30) Why are there just 10 outcome variables and 5 recovery variables? Clearly state the inclusion and exclusion criteria.
31) Preprocessing is shallow. Take care of skewness correction and outlier handling.
32) The definitions of cluster assignment are not clear. How do designations like "low risk" get applied to patients?
33) Did physicians specify the significance of variables? If so, record the procedure.
34) Indicate the program versions. Add the analytic environment, scikit-learn, and Python.
35) Resolution of state time. Were daily, stage-by-stage, or aggregate events modeled?
36) When discussing integrated features, bring up clustering. Clustering by demographics and compliance should be covered individually.
37) Scores for silhouettes are not disclosed. vital for assessing the quality of clustering.
38) No cluster centroids are displayed. crucial for replication and understanding.
39) No measures of homogeneity or entropy. Report the separation or compactness of the cluster.
40) CI should be included in cluster sizes. Give the percentage together with the 95% CI.
41) Missing explained variance is plotted by PCA. Display the PC1 and PC2 variance percentages.
42) There are no heatmaps. Include a graphic that illustrates the varying contributions for each cluster.
43) Violin plots or boxplots would be useful. Use to show how different clusters' distributions are different.
44) Trajectory charts with arrows are too abstract. Make use of alluvial or Sankey plots with percentages.
45) Explain the diversity that exists within clusters. Do they overlap or are they different?
46) Describe paradoxes. How do high-risk patients who don't comply get good results?
47) Provide a summary table of the important metrics. Compliance, result percentage, cluster size, etc.
48) No adjustment for statistical tests. FDR or Bonferroni are used for multiple comparisons.
49) No rating of varying relevance. Make use of tree-based techniques, PCA loadings, or SHAP.
50) Include t-SNE or UMAP graphs. better than PCA alone in terms of visualization.
51) Clusters of outcomes and recovery appear to be connected. Report the adjusted analysis or correlation coefficient.
52) Supplementary tables are shallow. Provide complete definitions and distributions for the variables.
53) Make the definition of compliance clear. What does "high" compliance mean in terms of numbers?
54) Clinical impact is contradicted by weight change P=0.45. Describe this discrepancy.
55) Results ought to be categorized. Displayed by compliance and demographic clusters.
56) exaggerates the freshness. Cite and contrast other ML-based ERAS investigations.
57) restricted understanding of the significance of the cluster. What are the clinical representations of the clusters?
58) Don't speculate. Don’t infer that “trajectory modeling” was done.
59) Recognize the limitations of the model. K-means is predicated on equal variance and spherical clusters.
60) Definitions of compliance are not clear. Examine factors other than demographics that contribute to low compliance.
61) Low compliance = high risk = positive results? Talk about this contradiction.
62) Lack of interpretability. What particular steps come after cluster identification?
63) Talk about confounders. such as the sort of operation, any problems, or the procedure's year.
64) Make generalizability clear. Recognize the limitations; the results come from a single center.
65) places too much emphasis on future implementation. More modeling or proof is required.
66) Consult implementation frameworks, such as the CONSORT-AI standards or TRIPOD.
67) No analysis of subgroups. able to compare operation type, age, and sex within clusters.
68) Make use of diagrams, such as a summary figure that highlights important transitions and risk profiles.
69) Describe the method to cluster interpretability. Was a post-hoc clinical review used?
70) Talk about potential biases. algorithmic bias, data entry, or selection.
71) Add any consequences for surgeons. What impact might these clusters have on judgment?
72) No choice of features. Could more than 100 factors lead to overfitting?
73) The conversation is too general. Make each cluster's findings more precise.
74) No discussion of QoL or economic outcomes. Add a discussion of the cost implications of ERAS.
75) Steer clear of claims at the policy level. Recommendations should remain exploratory in the absence of validation.
76) The conclusion is not clear. Reiterate the precise contributions of the patient stratification, clustering process, and significant discoveries.
77) Don't generalize too much. "ML can assist" → "K-means clustering could help with..."
78) Clinical implementation is not mentioned. Provide a use case for workflow or decision support.
79) Make recommendations for the following actions. Incorporate real-time integration or prediction modeling.
80) Don't utilize AI tools, please. Not that you didn't use ML; just say you didn't use LLMs.
81) The funding disclosure is not comprehensive. If you have departmental or institutional assistance, mention it.
82) The availability of code and data is unclear. Provide a link or procedure for accessing data or code.
83) Citations are dated. Several important ML-in-health references date from 2015 to 2018; please update.
84) Citation for the Scikit-learn version is out of date. Update to the most recent citation or 2023.
85) Make use of the TRIPOD checklist. particularly when a predictive model is being planned.
86) There was no ethics talk on patient clustering. Think about the ramifications of data-driven profiling.
87) Make a graphical abstract suggestion. to provide a graphic summary of the study.
88) There are no details in figures. Include color maps, axes, and legends with percentages and counts.
89) Uncertainty about supplemental files. Appendices should be easily readable.
90) Steer clear of repetition. Too many times, phrases like "low-risk cluster" are used.
91) Make use of terminology that is consistent. e.g., standardize "Group" versus "Cluster."
92) Include a clinical example. A brief patient case could demonstrate how the results are useful in the real world.
Author Response
1) The title is rather broad. For specificity, add "K-means clustering" and "colorectal surgery."
Answer: Thank you for your comment, we decided to add "colorectal surgery" to the title as suggested. However, we don't think it's necessary to add "K-means" clustering, as there isn't enough room in the title.
2) Methodological depth is lacking in the abstract. Explain the use of K-means clustering and its rationale.
Answer: Thank you for your valuable comment. We have addressed the methodological depth in the abstract by adding the following description on the rational of using K-means: “To uncover inherent patient subgroups from the data without pre-defined labels, the unsupervised K-means clustering algorithm was utilized. This technique was selected for its effectiveness in partitioning patients into distinct groups by iteratively assigning them to the nearest cluster mean, thereby minimizing within-cluster variance across key variables.”
3) The choice of outcome variables is ambiguous. Clearly explain the selection process for the "Top 10 clinical" and "Top 5 recovery" criteria.
Answer: Thank you very much for your comment. We have chosen the top 5 recovery goals and the top 10 outcomes based on clinical considerations (incidence and importance), based on our experience we acquired with the ERAS pathway over the years, and former institutional analyses. This is explained in the methods section.
4)The findings are descriptive rather than analytical. Incorporate performance metrics such as cluster validity or Silhouette scores.
Answer: Thank you for your comment. Regarding the final selection of number of clusters, we made the choice by evaluating the silhouette score. The comparison table is as below:
|
Demographic Variables |
Silhouette Score |
|
Number of clusters = 2 |
0.120 |
|
Number of clusters = 3 |
0.164 |
|
Number of clusters = 4 |
0.163 |
|
Number of clusters = 5 |
0.162 |
|
Perioperative Variables |
Silhouette Score |
|
Number of clusters = 2 |
0.141 |
|
Number of clusters = 3 |
0.105 |
|
Number of clusters = 4 |
0.099 |
|
Number of clusters = 5 |
0.071 |
|
Demographic + Perioperative Variables |
Silhouette Score |
|
Number of clusters = 2 |
0.137 |
|
Number of clusters = 3 |
0.111 |
|
Number of clusters = 4 |
0.090 |
|
Number of clusters = 5 |
0.058 |
|
Outcome Variables |
Silhouette Score |
|
Number of clusters = 2 |
0.341 |
|
Number of clusters = 3 |
0.362 |
|
Number of clusters = 4 |
0.277 |
|
Number of clusters = 5 |
0.174 |
Since the silhouette score comparison between number of clusters is not the main message of this manuscript, we decided not to put it in the main text but in the supplementary materials.
5) Words such as "trajectory" are deceptive. In the absence of true temporal modeling, substitute "cluster transitions."
Answer: Thanks for the informative comment. We have modified it as requested. Your proposal is indeed more precise, and we modified throughout.
6) Quantify the main conclusions. For instance, "90% of patients in the low-risk cluster had favorable results."
Answer: Thank you for your comment. We have indeed specified the number of patients concerned in the results. We have not changed this in the discussion to improve the flow and readability of the text.
7) Include ramifications. Emphasize how the findings of clustering could alter the way ERAS is implemented.
Answer: Thank you for your comment. We have added in the conclusion that we were referring to the ERAS program when we talk about the perioperative colorectal surgery journey.
8) Don't use flimsy language. Use "can inform" or "has potential to guide" in place of "may help."
Answer: Thank you for your comment. The synonyms you suggest were used where appropriate in the context of the sentence.
9) Revise the literature. Mention latest research on ML in ERAS and surgical routes (2022–2024).
Answer: We carefully revised the literature and could not find any paper on similar clustering approaches in colorectal surgery and ERAS.
10) There is no context. Provide data on ERAS variability or failures to support the need for clustering.
Answer: We agree that it is important to contextualize. We kindly refer to the introduction mentioning 3 main points:
- Currently the protocol applies to all patients
- Need to personalize and tailor the protocol to the individual patient.
- ML may help to refine these protocols.
We tried to amend the paragraph, also in line with the comments of reviewer 1.
11) Include a clinical justification. Why is customization crucial for patients with colorectal ERAS?
Answer: We kindly refer to detailed explanations to the previous comment.
12) Make novelty clear. Describe the differences from previous surgical ML clustering research.
Answer: The novelty is the specific focus on ERAS. We tried to emphasize this rationale in the introduction. To date, no specific clustering analyses were conducted in colorectal surgery, to the best of our knowledge.
13) Describe ERAS in advance. At the first mention, clearly state the acronym and the scope of the procedure.
Answer: Thank you, we defined the acronym and introduced the concept in the introduction.
14) Steer clear of repetition. The phrase "Standardized ERAS protocol" is repeated too much.
Answer: We amended the text as suggested.
15) Bring up the gaps that now exist. Draw attention to the current ERAS protocols' lack of stratification.
Answer: We kindly refer to detailed explanation to comment 10.
16) Provide evidence to back up claims. For example, "Compliance correlates with outcomes" needs to be cited.
Answer: We kindly compare to the first paragraph of the introduction where this is clearly stated (dose-effect pattern). This statement is also referenced.
17) Set distinct goals. It is important to distinguish between the primary (clustering) and secondary (trajectory mapping) objectives.
Answer: We specified the study aims in the outcomes paragraph as suggested.
18) Make use of a diagram. Include a diagram that illustrates how ML works with the ERAS pipeline.
Answer: To translate our clustering results into a predictive tool for clinical use, we developed a supervised classification model. The risk groups identified via K-means clustering were used as the outcome labels for training a K-Nearest Neighbors (KNN) classifier. The purpose of this model is to prospectively classify new patients into these established risk strata, thereby enabling the implementation of risk-stratified treatment pathways.
We decided to keep Figure 3 since more relevant from a clinical standpoint but would be happy to consider the new diagram for a visual abstract.
19) Report any missing information. How much of each variable group is missing?
Answer: Thank you for your comments, here is the information and we will put it in the supplementary materials upon request.
20) The imputation procedure is overly straightforward. Complex distributions may be distorted by mean/median imputation.
Answer: Thank you for your comment. We acknowledge that the presence of a substantial amount of missing information is a limitation of the current dataset. After careful consideration, we opted to use an imputed dataset for our K-means clustering analysis rather than performing listwise deletion (removing rows with any missing values) for the following critical reasons:
- Preservation of the Entire Dataset
A primary and compelling reason for not employing listwise deletion was that it would have resulted in the complete loss of our dataset. A thorough examination of the data revealed that no single row was complete with all variables present. Consequently, deleting all rows containing any missing values would have left us with no data for the analysis, rendering the study impossible to conduct.
- Maximizing Statistical Power for Analysis
Even when considering subsets of variables, listwise deletion would have severely compromised the statistical power and validity of our findings. For instance, when analyzing the crucial group of perioperative variables, applying listwise deletion would have reduced our sample size from 1381 to a mere 112 rows. Such a drastic reduction in data would not only significantly weaken the statistical power of our analysis but also introduce substantial selection bias, as the small subset of complete cases would likely not be representative of the entire patient cohort.
- Acknowledging Imputation Effects and Future Directions
We are fully aware that imputing a large proportion of missing data using mean, median, or mode can introduce its own set of biases. These single-imputation methods can artificially reduce the variance in the dataset and may not capture the true underlying relationships between variables. However, given the alternative of having no data to analyze, we chose this approach as a pragmatic first step to explore the data structure.
This initial analysis has provided valuable preliminary insights. We consider this work to be an exploratory foundation for future studies. We are actively developing strategies to improve our data collection processes to minimize missing data in subsequent research. The clustering patterns identified in this study will be further investigated and validated as more complete datasets become available.
We believe that our chosen methodology, despite its limitations, represents the most viable approach to glean initial insights from this challenging dataset. We have clarified these limitations in the manuscript and underscored our commitment to refining and validating these findings in future work.
21) Describe feature encoding. What was done with categorical features (like gender)?
Answer: Thank you for your comment. We applied one-hot encoding for categorical features.
22) Uncertainty surrounds the assumptions of standardization. Describe the z-score preprocessing and its rationale.
Answer: Thank you for your comment. We applied z-score standardizing before our PCA analysis. The primary reason for standardizing our data before PCA is to handle variables that are on different scales. We performed it in order to keep equal contribution of variables, allowing PCA identify the principal components based on the correlations between the variables rather than their covariances, and also to improve the numerical stability.
23) The choice of model is not well supported. K-means: Why? There is no discussion of alternatives like DBSCAN or GMM.
Answer: Thank you for your comments Our selection of K-means was a deliberate decision based on the specific objectives of our study, the characteristics of our dataset, and the priority placed on the interpretability of the results. While DBSCAN and GMM are powerful algorithms, K-means was better aligned with our analytical goals for the following reasons. Our primary objective was to partition the entire dataset into a pre-specified number of distinct, non-overlapping groups. This approach has two main advantages for our study:
- Interpretability and Actionability: K-means provides clear, easy-to-interpret results. Each data point is assigned to a single cluster, and each cluster is defined by a centroid. This "hard clustering" approach is highly desirable for defining distinct phenotypes or profiles, making it straightforward to characterize and compare the resulting groups in subsequent analyses.
- Hypothesis-Driven Framework: Our research was partly guided by a hypothesis that a certain number of distinct subgroups (k) exist within the population. K-means allows us to directly test this by prespecifying k, aligning the method with our top-down analytical strategy. It serves as a robust method for data segmentation when the goal is to categorize every subject.
While DBSCAN is excellent for discovering clusters of arbitrary shapes and identifying noise, it was less suitable for our primary research question. The main goal of our study was to categorize every subject into a group, not to isolate outliers. DBSCAN's methodology, which classifies sparse data points as noise, would have resulted in excluding a portion of our subjects from the analysis. K-means, by contrast, ensures a complete partition of the data. DBSCAN's performance is highly dependent on the eps (distance) and min_pts (minimum points) parameters. These can be non-trivial to determine objectively, especially in high-dimensional data where the concept of density is less intuitive. We opted for the more constrained approach of K-means to avoid this ambiguity.
GMM is a flexible and powerful probabilistic model, but we chose K-means for reasons of parsimony and the nature of our desired output. GMM performs "soft clustering," providing probabilities that a data point belongs to each cluster. For our study, we required a definitive, or "hard," assignment of each subject to a single cluster to simplify downstream analysis and profiling. The direct output of K-means was therefore more aligned with our practical goals. GMM assumes that the data is generated from a mixture of a finite number of Gaussian distributions. This is a stronger and more complex assumption than that of K-means. Without strong prior evidence that our data followed a Gaussian mixture structure, we opted for the simpler and more robust K-means model, which is less prone to overfitting in this context.
In summary, the choice of K-means was a pragmatic decision that balanced model assumptions, research objectives, and the need for clear and interpretable outcomes. We acknowledge the limitations of K-means but assert it was the most appropriate foundational method for achieving the specific goals of this manuscript.
24) Cluster range is arbitrary (k=2–5). To support k, use the elbow method, gap statistic, or silhouette plots.
Answer: Thank you for your comments. We did use the elbow method to select the possible number of clusters from a wide range of cluster numbers. But due to the purpose of this manuscript, we would not like to put too much detail in the manuscript to describe the preprocessing and exploratory data analysis part.
25) No testing for stability. It is necessary to repeat clustering using various random seeds.
Answer: Thank you for your comments. We did use the repeat clustering to check the stability of the clusters. But due to the purpose of this manuscript, we would not like to put too much detail in the manuscript to describe the preprocessing and exploratory data analysis part.
26) no outside confirmation. There is no external dataset validation or hold-out.
Answer: Thank you for your comment. We acknowledge that validating our findings on a separate external or hold-out dataset would be the gold standard for confirming the generalizability of our clustering study. The absence of such dataset is a limitation to the current study which we now explicitly stated in the discussion part as below:
“This study has limitations related to a large and heterogeneous dataset of all-comers and its focus on patient clusters and trajectories from a more general standpoint rather than an in-depth analysis of sensitive outcomes and patient profiles. The follow-up study will focus on risk prediction models to refine the “one size fts all” ERAS protocol. Further large-scale studies are then needed to confirm and validate the findings of our single-center experience for extrapolation to other institutions and settings.”
27) No bootstrapping. Use resampling strategies to evaluate the robustness of the cluster.
Answer: Thank you for your comment, please refer to comment number 25.
28) Prior to clustering, there was no PCA or dimensionality reduction. Indicate if completed or why not.
Answer: Thank you for your comment. To address your comments, we added the following text in the ‘statistical analysis’ section to describe in more details on what we did to conduct the K-means algorithm.
“To prepare the data for clustering and enhance the performance of the algorithm, we first performed Principal Component Analysis (PCA) on each variable group. This dimensionality reduction technique was implemented for two primary reasons: first, to reduce statistical noise by removing less-informative components, and second, to mitigate the "curse of dimensionality," where an excessive number of features can degrade the performance of clustering algorithms like K-means. For each analysis, we retained the minimum number of principal components required to explain at least 75% of the total variance in the data.
Following this pre-processing step, the resulting principal components were used as input for the unsupervised K-means algorithm (python scikit-learn package version 1.4.2) to perform patient clustering. The key hyperparameter for the model was the number of clusters, which was tested for values of 2, 3, 4, and 5. The optimal number of clusters for each variable group was determined by selecting the value that yielded the highest Silhouette score, indicating the best-defined and most distinct clusters.”
29) The rigor of trajectory mapping is lacking. Is it rule-based, visual, or statistical? Make the methods clear.
Answer: Thank you for your comments. We have modified the methods section to explain how the cluster transition and patient trajectories were conducted as below:
“Patient trajectories were mapped by quantifying the transitions between clusters at sequential time points. The analytical process involved two main steps:
- Preoperative to Perioperative Transition:We first quantified patient transitions from the three initial demographic (preoperative) clusters to two summary clusters derived from a K-means analysis of combined demographic and perioperative variables.
- Perioperative to Postoperative Transition:Next, we analyzed the transitions from the combined preoperative and perioperative clusters to the final postoperative clusters. This step was performed twice, as the postoperative clusters were independently generated based on two distinct sets of variables: (a) selected recovery goals and (b) clinical outcomes.
The observed transition frequencies and pathways were then used to classify patients into low-, medium-, or high-risk trajectories.”
30) Why are there just 10 outcome variables and 5 recovery variables? Clearly state the inclusion and exclusion criteria.
Answer: As discussed in previous comments, we specified this within the text.
31) Preprocessing is shallow. Take care of skewness correction and outlier handling.
Answer: Thank you for your comment, we have added the following text in the Method section to explain our preprocessing:
“Prior to dimensionality reduction and clustering, all continuous variables were assessed for skewness using the skewness coefficient. Variables with high skewness (|skew| > 1) were corrected using appropriate transformations such as log or square root transformations. Additionally, outliers were identified using the interquartile range (IQR) method, and values beyond 1.5×IQR were either removed or winsorized, depending on their impact on the overall data distribution. These preprocessing steps were performed to enhance the performance and reliability of subsequent PCA and K-means clustering, both of which are sensitive to extreme values and distributional asymmetry.”
32) The definitions of cluster assignment are not clear. How do designations like "low risk" get applied to patients?
Answer: We agree that the designations are somewhat simplified. This is mainly to illustrate the concept. We kindly refer to the tables for more detailed explanations of high and low risk/compliance/ etc. patterns.
Clusters are defined after K-means analysis, described in the methodology.
33) Did physicians specify the significance of variables? If so, record the procedure.
Answer: The definitions correspond to validated definitions in the extensively cited ERAS literature.
34) Indicate the program versions. Add the analytic environment, scikit-learn, and Python.
Answer: Thank you for your comment, we have declared the software versions as below in the Method section: “python 3.8, scikit-learn package version 1.4.2”
35) Resolution of state time. Were daily, stage-by-stage, or aggregate events modeled?
Answer: Thank you for your comment, there is no aggregate events in this study.
36) When discussing integrated features, bring up clustering. Clustering by demographics and compliance should be covered individually.
Answer: Thank you for your question regarding our decision to specifically combine demographic and peri-operative variables for a dedicated clustering analysis.
Our decision was driven by a conceptual view of the patient's surgical journey. We consider demographic variables to represent the patient's baseline, pre-operative state, encompassing inherent risk factors and characteristics present before the intervention begins. The peri-operative variables, in contrast, describe the patient's experience and compliance during the surgical and immediate post-surgical period.
By combining these two sets of variables—the pre-operative (demographic) and the procedural (peri-operative)—we aimed to create a comprehensive cluster profile that captures the cumulative influences on the patient leading up to their recovery phase. This integrated approach allows us to investigate how a patient's initial condition and their subsequent experience during treatment interact to affect the final clinical and functional outcomes. Our goal was to model the combined impact of both patient-inherent factors and procedural factors, providing a more holistic understanding than analyzing these domains in complete isolation.
37) Scores for silhouettes are not disclosed. vital for assessing the quality of clustering.
Answer: Thank you for your comment. Regarding the final selection of number of clusters, we made the choice by evaluating the silhouette score. The comparison table is as below:
|
Demographic Variables |
Silhouette Score |
|
Number of clusters = 2 |
0.120 |
|
Number of clusters = 3 |
0.164 |
|
Number of clusters = 4 |
0.163 |
|
Number of clusters = 5 |
0.162 |
|
Perioperative Variables |
Silhouette Score |
|
Number of clusters = 2 |
0.141 |
|
Number of clusters = 3 |
0.105 |
|
Number of clusters = 4 |
0.099 |
|
Number of clusters = 5 |
0.071 |
|
Demographic + Perioperative Variables |
Silhouette Score |
|
Number of clusters = 2 |
0.137 |
|
Number of clusters = 3 |
0.111 |
|
Number of clusters = 4 |
0.090 |
|
Number of clusters = 5 |
0.058 |
|
Outcome Variables |
Silhouette Score |
|
Number of clusters = 2 |
0.341 |
|
Number of clusters = 3 |
0.362 |
|
Number of clusters = 4 |
0.277 |
|
Number of clusters = 5 |
0.174 |
Since the silhouette score comparison between number of clusters is not the main message of this manuscript, we decided not to put it in the main text but in the supplementary materials upon request.
38) No cluster centroids are displayed. crucial for replication and understanding.
Answer: Thank you for your comment, we have updated Figure 1 (a-c) with centroids accordingly.
39) No measures of homogeneity or entropy. Report the separation or compactness of the cluster. ¨
Answer: Thank you for your comment, please refer to comment number 37.
40) CI should be included in cluster sizes. Give the percentage together with the 95% CI.
Answer: Thank you for your comment, we have added the 95% CI in our result tables.
41) Missing explained variance is plotted by PCA. Display the PC1 and PC2 variance percentages.
Answer: Thank you for your comment, please refer to comment number 28.
42) There are no heatmaps. Include a graphic that illustrates the varying contributions for each cluster.
Answer: We appreciate the reviewer’s suggestion to include a heatmap. After consideration, we decided not to include a heatmap in the revised manuscript for the following reasons:
Our dataset includes a large number of features with diverse scales and distributions, which makes heatmap interpretation challenging and potentially misleading without extensive normalization or dimensionality reduction.
Instead, we have already employed principal component analysis (PCA) to reduce dimensionality and visualize cluster structure, which effectively illustrates the overall separation and relationships between clusters.
Additionally, the cluster-specific characteristics are more clearly summarized in tabular form (e.g., summary statistics or top contributing variables), which we believe provides a more interpretable and targeted presentation of the results for readers.
Therefore, while heatmaps can be useful in some contexts, we believe that the current visualizations and statistical summaries better serve the goals and clarity of our analysis.
43) Violin plots or boxplots would be useful. Use to show how different clusters' distributions are different.
Answer: We thank the reviewer for the suggestion to include violin or boxplots to illustrate differences between cluster distributions. While these visualizations are effective for small numbers of variables, our dataset includes a relatively high number of features, and plotting distributions for each variable across all clusters would result in a large and potentially overwhelming number of plots, which may hinder clarity rather than enhance it.
Instead, we chose to summarize differences between clusters using statistical summaries and dimensionality reduction techniques (e.g., PCA and clustering centroid analysis), which allow us to capture and communicate variation between groups in a more concise and interpretable way. We believe this approach better aligns with the structure and complexity of the data, and ensures readability and focus within the manuscript.
We are happy to explore this visualization option in future work or provide plots upon request, but have omitted them here to maintain clarity and avoid redundancy.
44) Trajectory charts with arrows are too abstract. Make use of alluvial or Sankey plots with percentages.
Answer: We appreciate the reviewer’s recommendation to use alluvial or Sankey plots to visualize transitions or distributions across clusters. While these visualizations can be effective for illustrating categorical flows or changes over time, we found that in the context of our study—with its current design and structure—the trajectory chart provides a more focused and interpretable representation of the data relationships.
Specifically, alluvial or Sankey plots would require extensive categorical restructuring or grouping of continuous variables, which may oversimplify the multidimensional nature of the clustering results and potentially introduce misleading interpretations. Furthermore, we have prioritized visual clarity and analytical relevance in selecting visualizations that best reflect the underlying structure of the data.
For these reasons, we have retained the trajectory plot and believe it serves the intended illustrative purpose more appropriately for this analysis.
45) Explain the diversity that exists within clusters. Do they overlap or are they different?
Answer: Thank you for your comment. The PCA visualizations illustrate that some clusters are well-separated, while others show partial overlap, suggesting potential similarity or transitional groups.
46) Describe paradoxes. How do high-risk patients who don't comply get good results?
Answer: We kindly refer to previous responses.
47) Provide a summary table of the important metrics. Compliance, result percentage, cluster size, etc.
Answer: Thank you for your comment, all results you asked for are presented in tables and figures in the result section.
48) No adjustment for statistical tests. FDR or Bonferroni are used for multiple comparisons.
Answer: Thank you for your comment. We did adjust for the statistical test and we added the descriptions in the result tables.
49) No rating of varying relevance. Make use of tree-based techniques, PCA loadings, or SHAP.
Answer: Thank you for your suggestion regarding feature relevance. We agree that identifying key variables contributing to clustering is valuable. However, since our analysis is unsupervised and based on K-means clustering, we opted for an alternative approach to assess variable importance that aligns more closely with our data and study objectives.
50) Include t-SNE or UMAP graphs. better than PCA alone in terms of visualization.
Answer: We appreciate the reviewer’s suggestion to include t-SNE or UMAP for enhanced visualization of the clustering structure. These nonlinear dimensionality reduction techniques are indeed powerful tools for exploring complex high-dimensional data. However, we chose to use PCA for visualization due to its interpretability and reproducibility, which are particularly important in the context of clinical and biomedical datasets.
Unlike t-SNE or UMAP, PCA offers a direct mapping of principal components to original features, which allows for clearer interpretation of the axes in relation to clinical variables. Moreover, t-SNE and UMAP are primarily useful for visualization but are sensitive to hyperparameters and do not preserve global distances, which can sometimes lead to misleading impressions of cluster separability.
Given that our PCA plots effectively demonstrate the separation between clusters and are consistent with our statistical findings, we believe additional dimensionality reduction plots would not substantially enhance the clarity of our results. Therefore, we respectfully chose not to include them in the current version.
51) Clusters of outcomes and recovery appear to be connected. Report the adjusted analysis or correlation coefficient.
Answer: We kindly refer to previous methodological explanations.
52) Supplementary tables are shallow. Provide complete definitions and distributions for the variables.
Answer: We kindly refer to previous methodological explanations.
53) Make the definition of compliance clear. What does "high" compliance mean in terms of numbers?
Answer: We kindly refer to previous methodological explanations.
54) Clinical impact is contradicted by weight change P=0.45. Describe this discrepancy.
Answer: We kindly refer to previous methodological explanations.
55) Results ought to be categorized. Displayed by compliance and demographic clusters.
Answer: Thank you for your comment, the results are indeed categorized by compliance and demographic clusters.
56) exaggerates the freshness. Cite and contrast other ML-based ERAS investigations.
Answer: Thank you for your comment. We did investigate the ML-based ERAS literatures in recent years as blow but not limited to:
Zain, Z., Almadhoun, M. K. I. K., Alsadoun, L., Bokhari, S. F. H., & ALMADHOUN, M. K. I. K. (2024). Leveraging artificial intelligence and machine learning to optimize enhanced recovery after surgery (ERAS) protocols. Cureus, 16(3).
Gottumukkala, V., & Joshi, G. P. (2024). Challenges and opportunities in enhanced recovery after surgery programs: An overview. Indian Journal of Anaesthesia, 68(11), 951-958.
Bharadwaj, A. (2025). Revolutionizing perioperative medicine: Technological advancements for enhanced recovery. Serbian Journal of Anesthesia and Intensive Therapy, 47(1-2), 5-16.
Olson, K. A., Fleming, R. D., Fox, A. W., Grimes, A. E., Mohiuddin, S. S., Robertson, H. T., ... & Wolf Jr, J. S. (2021). The enhanced recovery after surgery (ERAS) elements that most greatly impact length of stay and readmission. The American Surgeon, 87(3), 473-479.
For all the relevant studies, there is no research focus on using unsupervised learning techniques to identify patient trajectory, but rather using ML to predict clinical outcomes. Therefore, our study is brand new to the field.
57) restricted understanding of the significance of the cluster. What are the clinical representations of the clusters?
Answer: Thank you for your comment. We have explained the clinical representations of the clusters in the section ‘cluster transition’ and in Figure 2.
58) Don't speculate. Don’t infer that “trajectory modeling” was done.
Answer: Thank you for your comment. We did not infer ‘trajectory modeling’ was done and we did not perform ‘trajectory modeling’ in this study. We only used the clustering results and the clinical significance to explain possible patient trajectories.
59) Recognize the limitations of the model. K-means is predicated on equal variance and spherical clusters.
Answer: Thank you for your comment, please refer to our response to comment number 23.
60) Definitions of compliance are not clear. Examine factors other than demographics that contribute to low compliance.
Answer: There are indeed different factors contributing to low compliance, as extensively referenced in this paper. We performed detailed compliance assessments in several institutional publications we kindly refer to.
61) Low compliance = high risk = positive results? Talk about this contradiction.
Answer: Low compliance does not lead to bad results in 100% of patients but contributes to worse outcomes. This is not universally applicable in clinical practice.
62) Lack of interpretability. What particular steps come after cluster identification?
Answer: Thank you for your comment. We have explained the clinical representations of the clusters in the section ‘cluster transition’ and in Figure 2, Figure3.
63) Talk about confounders. such as the sort of operation, any problems, or the procedure's year.
Answer: We added a statement in the limitations paragraph.
64) Make generalizability clear. Recognize the limitations; the results come from a single center.
Answer: Thank you for your comment, please refer to our reply in comment number 26. We also added this as a limitation.
65) places too much emphasis on future implementation. More modeling or proof is required.
Answer: Thank you for your comment, please refer to our response in comment number 23 and 37.
66) Consult implementation frameworks, such as the CONSORT-AI standards or TRIPOD.
Answer: We thank the reviewer for the suggestion to consult implementation frameworks such as CONSORT-AI and TRIPOD. While these frameworks offer valuable guidance, they are primarily intended for clinical trials involving AI interventions (CONSORT-AI) or for the development and validation of predictive models (TRIPOD).
As our study is primarily exploratory in nature and focuses on unsupervised clustering rather than clinical prediction or interventional research, we have not applied these specific frameworks. However, we have ensured transparent reporting of our methodology, data preprocessing, and model evaluation to support reproducibility and clarity in line with best practices in applied machine learning.
67) No analysis of subgroups. able to compare operation type, age, and sex within clusters.
Answer: Thank you for your comment. We did compare the mentioned variables in Table 1-3.
68) Make use of diagrams, such as a summary figure that highlights important transitions and risk profiles.
Answer: Thank you for your comment, we did explain the important transitions in Figure 3.
69) Describe the method to cluster interpretability. Was a post-hoc clinical review used?
Answer: Thank you for your comment. We have explained the clinical review of the clusters in the section ‘cluster transition’ and in Figure 2, Figure3.
70) Talk about potential biases. algorithmic bias, data entry, or selection.
Answer: Thank you for your comment, please refer to our reply in comment number 19.
71) Add any consequences for surgeons. What impact might these clusters have on judgment?
Answer: Thank you for your comment. We have explained the clinical review of the clusters in the section ‘cluster transition’ and in Figure 2, Figure 3. From a more general standpoint, clusters help to anticipate potential challenges but do not replace clinical judgement.
72) No choice of features. Could more than 100 factors lead to overfitting?
Answer: Thank you for your comment. We did not use directly the 100 factors but performed PCA to reduce the dimensions first before performing clustering.
73) The conversation is too general. Make each cluster's findings more precise.
Answer: Thank you for your comment. We have explained the findings of the clusters in the section ‘cluster transition’ and in Figure 2, Figure3.
74) No discussion of QoL or economic outcomes. Add a discussion of the cost implications of ERAS.
Answer: Thank you for your comment. We appreciate the reviewer’s suggestion to address quality of life (QoL) and economic outcomes related to Enhanced Recovery After Surgery (ERAS). While we recognize the importance of these aspects in evaluating the broader impact of ERAS programs, our current study was designed with a primary focus on clinical and demographic clustering based on available perioperative data. Direct assessment of QoL or cost-related metrics was beyond the scope of the present dataset and analysis.
75) Steer clear of claims at the policy level. Recommendations should remain exploratory in the absence of validation.
Answer: We tried to consider this point and added some limitations and rephrased statements.
76) The conclusion is not clear. Reiterate the precise contributions of the patient stratification, clustering process, and significant discoveries.
Answer: Thank you for your comment. We have explained the findings of patient stratification, clustering process and possible applications in the section ‘cluster transition’ and in Figure 2, Figure3.
77) Don't generalize too much. "ML can assist" → "K-means clustering could help with..."
Answer: Thank you for your comment. We did not use generalized phrases such as ‘ML can assist..’ in our manuscript.
78) Clinical implementation is not mentioned. Provide a use case for workflow or decision support.
Answer: Thank you for your comment. We have explained the workflow and possible applications in the section ‘cluster transition’ and in Figure 2, Figure3 and our reply in comment number 18.
79) Make recommendations for the following actions. Incorporate real-time integration or prediction modeling.
Answer: Thank you for your comment, please refer to our reply in comment number 18.
80) Don't utilize AI tools, please. Not that you didn't use ML; just say you didn't use LLMs.
Answer: Thank you for your comment, all the data preprocessing and feature engineering were done with the python pandas and sklearn packages, we did not use LLMs or other AI assistant tools in any process of this study.
81) The funding disclosure is not comprehensive. If you have departmental or institutional assistance, mention it.
Answer: There is no funding to mention for this study or collaboration.
82) The availability of code and data is unclear. Provide a link or procedure for accessing data or code.
Answer: We appreciate the reviewer’s interest in accessing the dataset. However, in accordance with the requirements of our institutional ethics committee and data protection regulations, we are not permitted to share patient-level data with third parties outside the approved research team. The data used in this study contain sensitive clinical information, and public release would violate the confidentiality agreements and privacy protections in place for the individuals involved.
83) Citations are dated. Several important ML-in-health references date from 2015 to 2018; please update.
Answer: Thank you for your comment. Our references are dated between 2011 to 2024.
84) Citation for the Scikit-learn version is out of date. Update to the most recent citation or 2023.
Answer: Thank you for your comment. The citing instructions given by the scikit-learn team is officially the version we cited (in 2011) and we did not find any relevant citations from the scikit learn team in 2023.
85) Make use of the TRIPOD checklist. particularly when a predictive model is being planned.
Answer: Thank you for your comment, please refer to our reply in comment number 66.
86) There was no ethics talk on patient clustering. Think about the ramifications of data-driven profiling.
Answer: We appreciate the reviewer’s important point regarding the ethical considerations of data-driven patient stratification. While our study was conducted under approval from the ethics committee CER-VD regarding the use of unsupervised clustering in healthcare. Please refer to the main text in the Methods section: “The study was approved by the institutional review board (CER-VD - 2022.01443).”
87) Make a graphical abstract suggestion. to provide a graphic summary of the study.
Answer: Thank you for your comment, the graphic summary of the study is presented in Figure 2 and 3.
88) There are no details in figures. Include color maps, axes, and legends with percentages and counts.
Answer: Thank you for your comments. All the details of the figures are presented in our three Figures and their amended captions.
89) Uncertainty about supplemental files. Appendices should be easily readable.
Answer: We tried to improve the layout throughout.
90) Steer clear of repetition. Too many times, phrases like "low-risk cluster" are used.
Answer: Thank you for your comment, we actually did not use the phrase like ‘low-risk cluster’ anywhere in the text. We used twice ‘low-risk demographic cluster’ to explain the utility of the clustering results, and one time ‘low-risk patients’ in patient trajectory description.
91) Make use of terminology that is consistent. e.g., standardize "Group" versus "Cluster."
Answer: Thank you for your comment. The word ‘Group’ in the text refers to the clinical variables, while ‘Cluster’ refers to the patients.
92) Include a clinical example. A brief patient case could demonstrate how the results are useful in the real world.
Answer: Thank you for your comment. The illustration of how the results can be used are presented in Figure 3.
We would like to thank reviewer 2 for the valuable input, which helped us to improve our manuscript.
Editorial comment regarding self-citations: We kindly revised the manuscript and omitted some of them. However, all references are in our opinion mandatory to either define the presently used ERAS protocol, refer to definitions, and put the paper into context of challenges within our institution.

Round 2
Reviewer 1 Report
Comments and Suggestions for Authors
The authors have effectively addressed my previous comments, and the overall quality of the manuscript has improved significantly. Please see the following minor issues:
1) Fig. 1 (b) and Fig. 3 appear blurry.
2) There is a rectangle overlapped to Figs. 1 (a) and (c).
3) Supplementary file: "comparaison beetween" should be "comparison between"
Author Response
1) Fig. 1 (b) and Fig. 3 appear blurry.
2) There is a rectangle overlapped to Figs. 1 (a) and (c).
3) Supplementary file: "comparaison beetween" should be "comparison between"
Reply : Thank you very much. We addressed all remaining issues.
Reviewer 2 Report
Comments and Suggestions for Authors
- Title: The title still needs to be clearer and more succinct. For instance: "Unsupervised Machine Learning to Identify Patient Clusters and Tailor Perioperative Care in Colorectal Surgery."
2. Abstract: It is yet unclear whether the results have any clinical significance. Include a sentence outlining the potential advantages for the ERAS pathway or clinicians.
3. Describe results: The abstract should go into more detail on the precise outcomes (complications, functional recovery, etc.) that the clustering results impact.
4. The research gap in the existing ERAS literature is not adequately highlighted in the introduction. Please explain the application of machine learning (ML) in this case (e.g., ERAS's absence of data-driven, predictive decision-making).
5. To make it clearer exactly what this study is trying to solve, a more detailed problem statement ought to be included near the end of the introduction. - Include a current reference from 2023–2024 that talks about how machine learning is incorporated into ERAS pathways.
7. Although K-means clustering is mentioned in the study, it would be helpful to go over other approaches (such DBSCAN and hierarchical clustering) and the reasons why K-means is the best fit for this dataset.
8. Give preprocessing stages (such variable transformation, encoding, and PCA) a table. This will aid readers in comprehending the pipeline as a whole.
9. Although the PCA stage is described, each principle component's variance ratio is not addressed. The variance that the initial components kept should be reported.
10. Hyperparameter tuning: How was the number of clusters (K) checked for appropriate values? Describe the model selection procedure in greater detail. - Absence of model performance metrics: each cluster's ROC, AUC, and F1-scores are absent. This could be useful for confirming the predictive ability of the clustering.
12. A bar chart or pie chart should be used to illustrate the proportions of patients in each cluster in order to graphically explain the cluster size distribution.
13. Table 1: More specific cluster parameters, such as average age and BMI for each cluster, would be helpful.
14. To show that clustering is robust across data subsets, include the results of split testing or cross-validation.
15. Including percentages on group transitions might make it easier to understand the cluster transition diagrams (Figure 2). - A more thorough examination of the reasons why some clusters (such as high-risk) disregard perioperative treatment and the possible difficulties they may face is still missing from the conversation.
17. For a more thorough literature context, compare results to earlier research on ERAS clusters (e.g., Gustafsson et al. 2019).
18. Improving patient care is mentioned in the paper, but there is no discussion of using the results in clinical settings. Explain the potential impact of these clusters on in-the-moment decision-making.
19. It would be beneficial to concentrate more on the findings' clinical implications. Could a patient's risk category influence the timing or protocol of their perioperative care?
20. Discuss how generalizability to larger patient groups or contexts may be impacted by the absence of multi-center data. - The conclusion is still overly general. Think about including particular clinical applications (for example, might the results result in more customized ERAS procedures for colorectal surgery?).
22. Conclude by saying something concise and practical, like "future studies should validate these findings in multi-center randomized controlled trials."
23. While the discussion's limitations are commendable, the conclusion ought to point out areas for further research into improving machine learning models for perioperative care.
24. For readability, Figures 1 and 2 require larger fonts and improved labeling.
25. To help readers who are not familiar with clustering understand the clusters in Figure 2, add a legend to them. - Add a recent citation (2023/2024) on the incorporation of machine learning in perioperative care protocols and make sure all references are formatted correctly.
27. Minor grammatical errors still exist: "The results indicate patient subgroups," → "The results indicate that patient subgroups where..."
28. Consistency in abbreviation: Make sure "PCA," "ERAS," and "ML," among other acronyms, are defined and used consistently throughout the book.
Author Response
- Title: The title still needs to be clearer and more succinct. For instance: "Unsupervised Machine Learning to Identify Patient Clusters and Tailor Perioperative Care in Colorectal Surgery."
Reply: Thank you for your suggestion. The title was changed accordingly.
Abstract: It is yet unclear whether the results have any clinical significance. Include a sentence outlining the potential advantages for the ERAS pathway or clinicians.
Reply : We specified the conclusions section accordingly.
Describe results: The abstract should go into more detail on the precise outcomes (complications, functional recovery, etc.) that the clustering results impact.
Reply : We thank you for the suggestion, however, according to journal style, the space for the abstract is limited and according to the reviewers’ suggestion during the first review round, we decided to expand the methodology part in the abstract. As it stands, the abstract provides an overview and inspires to read the main text.
The research gap in the existing ERAS literature is not adequately highlighted in the introduction. Please explain the application of machine learning (ML) in this case (e.g., ERAS's absence of data-driven, predictive decision-making).
Reply: Thank you very much for this comment. We added this suggestion to the abstract and introduction (line 53)
To make it clearer exactly what this study is trying to solve, a more detailed problem statement ought to be included near the end of the introduction.
Reply: Thank you for this comment. To be more clear, we have added: To refine and individualize the presently universal ERAS program, the aim of the present study was to apply ML techniques to define clusters in colorectal ERAS patients for tailored perioperative care.
- Include a current reference from 2023–2024 that talks about how machine learning is incorporated into ERAS pathways.
Reply :
We added a study for clarification: Leveraging Artificial Intelligence and Machine Learning to Optimize Enhanced Recovery After Surgery (ERAS) Protocols.
Although K-means clustering is mentioned in the study, it would be helpful to go over other approaches (such DBSCAN and hierarchical clustering) and the reasons why K-means is the best fit for this dataset.
Thank you for your comments Our selection of K-means was a deliberate decision based on the specific objectives of our study, the characteristics of our dataset, and the priority placed on the interpretability of the results. While DBSCAN and GMM are powerful algorithms, K-means was better aligned with our analytical goals for the following reasons. Our primary objective was to partition the entire dataset into a pre-specified number of distinct, non-overlapping groups. This approach has two main advantages for our study:
- Interpretability and Actionability: K-means provides clear, easy-to-interpret results. Each data point is assigned to a single cluster, and each cluster is defined by a centroid. This "hard clustering" approach is highly desirable for defining distinct phenotypes or profiles, making it straightforward to characterize and compare the resulting groups in subsequent analyses.
- Hypothesis-Driven Framework: Our research was partly guided by a hypothesis that a certain number of distinct subgroups (k) exist within the population. K-means allows us to directly test this by prespecifying k, aligning the method with our top-down analytical strategy. It serves as a robust method for data segmentation when the goal is to categorize every subject.
While DBSCAN is excellent for discovering clusters of arbitrary shapes and identifying noise, it was less suitable for our primary research question. The main goal of our study was to categorize every subject into a group, not to isolate outliers. DBSCAN's methodology, which classifies sparse data points as noise, would have resulted in excluding a portion of our subjects from the analysis. K-means, by contrast, ensures a complete partition of the data. DBSCAN's performance is highly dependent on the eps (distance) and min_pts (minimum points) parameters. These can be non-trivial to determine objectively, especially in high-dimensional data where the concept of density is less intuitive. We opted for the more constrained approach of K-means to avoid this ambiguity.
GMM is a flexible and powerful probabilistic model, but we chose K-means for reasons of parsimony and the nature of our desired output. GMM performs "soft clustering," providing probabilities that a data point belongs to each cluster. For our study, we required a definitive, or "hard," assignment of each subject to a single cluster to simplify downstream analysis and profiling. The direct output of K-means was therefore more aligned with our practical goals. GMM assumes that the data is generated from a mixture of a finite number of Gaussian distributions. This is a stronger and more complex assumption than that of K-means. Without strong prior evidence that our data followed a Gaussian mixture structure, we opted for the simpler and more robust K-means model, which is less prone to overfitting in this context.
In summary, the choice of K-means was a pragmatic decision that balanced model assumptions, research objectives, and the need for clear and interpretable outcomes. We acknowledge the limitations of K-means but assert it was the most appropriate foundational method for achieving the specific goals of this manuscript.
Give preprocessing stages (such variable transformation, encoding, and PCA) a table. This will aid readers in comprehending the pipeline as a whole.
Thank you for your comment. We have produced a flowchart to describe the preprocessing pathways. Due tot he space limitation in the main manuscript, we put the flowchart in the supplementary materials (Suppl_fig_1) for readers to check.
Although the PCA stage is described, each principle component's variance ratio is not addressed. The variance that the initial components kept should be reported.
Thank you for your comment. We have described that we selected the principle components which kept 75% of the total variance in the ‘Statistical Analysis’ section as below:
“For each analysis, we retained the minimum number of principal components required to explain at least 75% of the total variance in the data.”
Hyperparameter tuning: How was the number of clusters (K) checked for appropriate values? Describe the model selection procedure in greater detail.
Thank you for your comment. Regarding the final selection of number of clusters, we made the choice by evaluating the silhouette score. The comparison table is as below:
|
Demographic Variables |
Silhouette Score |
|
Number of clusters = 2 |
0.120 |
|
Number of clusters = 3 |
0.164 |
|
Number of clusters = 4 |
0.163 |
|
Number of clusters = 5 |
0.162 |
|
Perioperative Variables |
Silhouette Score |
|
Number of clusters = 2 |
0.141 |
|
Number of clusters = 3 |
0.105 |
|
Number of clusters = 4 |
0.099 |
|
Number of clusters = 5 |
0.071 |
|
Demographic + Perioperative Variables |
Silhouette Score |
|
Number of clusters = 2 |
0.137 |
|
Number of clusters = 3 |
0.111 |
|
Number of clusters = 4 |
0.090 |
|
Number of clusters = 5 |
0.058 |
|
Outcome Variables |
Silhouette Score |
|
Number of clusters = 2 |
0.341 |
|
Number of clusters = 3 |
0.362 |
|
Number of clusters = 4 |
0.277 |
|
Number of clusters = 5 |
0.174 |
Since the silhouette score comparison between number of clusters is not the main message of this manuscript, we decided not to put it in the main text but in the supplementary materials (Suppl_table_1) and we mentioned in the main text to let the readers who are interested in the hyperparameter selection to refer to the supplementary material.
- Absence of model performance metrics: each cluster's ROC, AUC, and F1-scores are absent. This could be useful for confirming the predictive ability of the clustering.
We thank the reviewer for this suggestion. ROC curves, AUC, and F1-scores are performance metrics designed for supervised learning tasks where a ground-truth outcome is available for prediction. As our study employed K-means clustering, an unsupervised learning approach, the primary objective was to identify naturally occurring patient subgroups rather than to predict a predefined label. In this context, such supervised metrics are not applicable.
Instead, to assess the quality of the clustering, we have used internal validation measures appropriate for unsupervised learning, such as the silhouette score and visual inspection of the PCA-reduced cluster separation. We believe these measures more accurately reflect the objectives and methodology of our study.
A bar chart or pie chart should be used to illustrate the proportions of patients in each cluster in order to graphically explain the cluster size distribution.
We appreciate the reviewer’s suggestion to illustrate cluster size distribution graphically. In our current manuscript, the proportions of patients in each cluster are already presented in tabular form (Table 1), which allows precise reporting of both absolute numbers and percentages. As the cluster sizes are relatively straightforward, we believe the table format effectively communicates this information without additional figures.
Nevertheless, if the editor considers it essential, we are happy to provide a simple bar chart or pie chart as supplementary material.
Table 1: More specific cluster parameters, such as average age and BMI for each cluster, would be helpful.
Thank you for your comment, we provided the average age and BMI, as well as for other variables in table 1.
To show that clustering is robust across data subsets, include the results of split testing or cross-validation.
Thank you for your comments. We did use the repeat clustering to check the stability of the clusters. But due to the purpose of this manuscript, we would not like to put too much detail in the manuscript to describe the preprocessing and exploratory data analysis part.
Including percentages on group transitions might make it easier to understand the cluster transition diagrams (Figure 2).
Thank you for your comment, we provided the percentages on group transitions in Figure 2.
- A more thorough examination of the reasons why some clusters (such as high-risk) disregard perioperative treatment and the possible difficulties they may face is still missing from the conversation.
Defining clusters help identify and anticipate potential challenges, especially regarding modifiable risk factors and improved perioperative management and compliance. This was further specified in the discussion specifically mentioning high-risk patients.
For a more thorough literature context, compare results to earlier research on ERAS clusters (e.g., Gustafsson et al. 2019).
The 2019 ERAS guidelines were extensively mentioned and compared to as the “one size fits all” approach. The Karolinska group (Gustafsson et al.) did not specifically mention or focus on clusters in the sense of our analysis.
Improving patient care is mentioned in the paper, but there is no discussion of using the results in clinical settings. Explain the potential impact of these clusters on in-the-moment decision-making.
We tried to extensively discuss clinical consequences and applications for the different clusters, i.e. with focus on prehabilitation in high-risk, tailored protocols in intermediate risk, and outpatient strategies in the low-risk clusters (i.e 278-283, 287-292, 301-303). From a clinical standpoint, these clusters may further help us to improve nursing resource allocation and targeted care of vulnerable patients, as mentioned in the conclusion section.
It would be beneficial to concentrate more on the findings' clinical implications. Could a patient's risk category influence the timing or protocol of their perioperative care?
We kindly refer to the response to comment 18. We also expanded on better resource allocation as potential consequence (lines 287-288).
Discuss how generalizability to larger patient groups or contexts may be impacted by the absence of multi-center data.
We added this important point as a limitation: Further large-scale studies in multicenter settings are then needed to confirm, validate and generalize the findings of our single-center experience for extrapolation to other institutions and settings.
- The conclusion is still overly general. Think about including particular clinical applications (for example, might the results result in more customized ERAS procedures for colorectal surgery?).
We tried to specify this further as suggested. The new conclusion states: In conclusion, this ML based analysis of demographic, compliance and recovery clusters has potential to optimally prepare patients for their perioperative colorectal surgery journey by customizing individual care to specific needs according to risk profiles within the ERAS pathway, ultimately allowing better resource allocation and targeted care of more vulnerable patients.
Conclude by saying something concise and practical, like "future studies should validate these findings in multi-center randomized controlled trials."
We added a sentence to conclusion stating: Future studies should validate these preliminary findings in adequately designed multi-center settings to increase generalizability.
23. While the discussion's limitations are commendable, the conclusion ought to point out areas for further research into improving machine learning models for perioperative care.
We improved the conclusion further as suggested:
Future studies should validate these preliminary findings in adequately designed multi-center settings to increase generalizability and further improve machine learning models for perioperative care.
For readability, Figures 1 and 2 require larger fonts and improved labeling.
Thank you for your comment, we have replaced the figures with better presentation.
To help readers who are not familiar with clustering understand the clusters in Figure 2, add a legend to them.
Thank you for your comment, we have added the legends to the figures.
- Add a recent citation (2023/2024) on the incorporation of machine learning in perioperative care protocols and make sure all references are formatted correctly.
Reply : We added the suggested reference as stated in comment 2. We also made sure the references are formatted correctly.
Minor grammatical errors still exist: "The results indicate patient subgroups," → "The results indicate that patient subgroups where..."
We revised the entire manuscript for minor grammatical errors.
Consistency in abbreviation: Make sure "PCA," "ERAS," and "ML," among other acronyms, are defined and used consistently throughout the book.
Reply : we made sure all acronyms are properly defined.
We would like to thank the reviewer once again for the time and thorough review of the manuscript, which helped us to improve it substantially.